# “Mortality, or not mortality, that is the question …”: How to Treat Removals in Tree Survival Analysis of Central European Managed Forests

**DOI:** 10.3390/plants13020248

**Published:** 2024-01-15

**Authors:** Paweł Lech, Agnieszka Kamińska

**Affiliations:** 1Department of Forest Resources Management, Forest Research Institute, Sękocin Stary, ul. Braci Leśnej 3, 05-090 Raszyn, Poland; 2Department of Geomatics, Forest Research Institute, Sękocin Stary, ul. Braci Leśnej 3, 05-090 Raszyn, Poland; a.kaminska@ibles.waw.pl

**Keywords:** tree mortality, managed stands, tree removals, thinning, sanitary cuttings

## Abstract

Tree mortality is an objective forest health criterion and is particularly suitable for long-term and large-scale studies of forest condition. However, it is impossible to determine actual tree mortality in Central European managed forests where trees are removed for various reasons. In this case, the only way to approximate tree mortality is to define the range in which it occurs. This can be carried out by including in the mortality calculations either dead trees that remain in the stand at the end of the assessment period or additionally trees that have been removed from the stand. We used data from the annual forest monitoring surveys in Poland from 2009 to 2022 for pine, spruce, oak and birch to perform a survival analysis in which we included all removals or sanitary cuttings either as censored or complete observations. The differences between the calculated mortality rates were significant, indicating the importance of how removals are treated in the analysis. To assess which method used for mortality calculation was more appropriate, we compared values for last recorded defoliation and severity of damage from live, dead and thinned or salvaged trees. For all species studied, significant differences were found between dead trees or trees removed by sanitation cuts and living trees or trees removed by thinning, suggesting that not only dead trees remaining in the forest, but also trees removed by sanitation cuts, should be considered when calculating mortality in managed stands. We also recommend the use of survival analysis in forest monitoring as a routine method for assessing the health of stands.

## 1. Introduction

Mortality is a fundamental process that determines population structure and dynamics. Together with health characteristics such as defoliation, crown transparency, severity of tree damage or reduction in increment, it is used as a criterion for assessing forest health, both in large-scale studies at continental [1,2,3,4], regional [5,6,7,8] and local scales [9]. It has also been the basis for the development of tree survival functions that have been widely used for growth modelling [10,11] and economic planning in forests [12,13].

The usefulness of tree mortality in the assessment of forest condition results from its objectivity, simplicity and accuracy of determination. It is an indicator that reflects all the processes occurring in the forest ecosystem, from the natural dynamics of the stand to the effects of all biotic, abiotic and anthropogenic stress factors. Relationships between tree mortality and other indicators of tree vigour have been described and documented in the literature: positive with tree defoliation [14,15,16,17,18] and negative with an increase in radial growth [16,19,20,21,22]. The inclusion of different environmental variables in the analyses makes it possible to determine the context of the mortality that occurs and to indicate its causes [8]. This makes tree mortality rate an extremely useful tool in forest health research. Supernatural mortality is usually associated with climate change and the resulting threats to the forest environment [23,24,25,26], as well as with the emergence of invasive alien species and pathogens in the context of globalisation processes [5,27].

Determining tree mortality in unmanaged forests is possible and simple, as neither dead nor living trees are removed. To find out the exact tree mortality rate, only an inventory of living trees at two points in time (the beginning and end) is required, because mortality is defined as the ratio of the difference between the number or volume of living trees at the beginning and end of the assessment period (not including ingrowing individuals) and the number or volume of living trees at the beginning of this period [28]. Unfortunately, there are only a few unmanaged forests in Europe, which occupy a relatively small area [29,30] and are usually located in places that are difficult to access. Nevertheless, the literature on tree mortality usually refers to forests that are restricted from direct human influence [31]. So how should the mortality rate of trees in managed forests, which are subject to silvicultural and protective measures such as thinning and sanitary cutting, be determined?

According to Dobbertin [32], it is impossible to determine the actual tree mortality rate in managed stands, as it is unlikely to be able to find out whether a removed tree was dead at the time of removal or whether it had died by the next survey or was still alive at that time, especially as surveys are carried out at regular intervals of 1, 5 or even 10 years [33,34]. This statement is basically correct, but is there no way to determine the mortality of trees at least approximately in managed stands or to indicate the range in which this mortality falls? This question also raises the issue of the meaning of the term “mortality”. If one takes the quoted definition of tree mortality [28] literally, all trees that are removed during the intermediate harvest should also be included in the calculations of tree mortality—they are all no longer alive. This approach can be supported by the phenomenon of self-thinning of stands, a natural process that indicates the presence of a barrier at a site that is able to maintain a certain number of stems with certain dimensions/biomass per unit area [35,36] or to grow with a certain stand sparsity [37]. Intermediate cuttings can be treated analogously to self-thinning, as their intensity is determined by the rules and relationships that govern the self-thinning process, which is one of the principles of modern forestry [38].

The inclusion of trees removed in intermediate cuts in the calculation of tree mortality in managed stands may seem controversial and is not mentioned in the literature on this topic. The prevailing approach is to include only trees that have been confirmed with certainty to have died between successive inventories [3,39,40]. In this case, however, all dead trees that were felled and removed from the stand, as well as those that were felled while still alive, although they would die by the next inventory, are excluded from the mortality calculations, as it is simply impossible to confirm with certainty the death of a removed tree if only a stump remains months or years after the tree was felled. Therefore, the calculated mortality rates are underestimated and do not reflect the actual health of the stand. For this reason, when calculating tree mortality in managed stands, trees that have been removed from the stand as sanitary cuttings are sometimes taken into account in addition to the dead trees remaining in the stand [41,42]. In this case, it is assumed that this calculation method provides a better approximation of the actual mortality in the stand. Dead, dying or severely damaged trees are usually subjected to intermediate cuttings. It is important to note, however, that sanitary cutting also removes healthy trees (i.e., trees in buffer zones around outbreak spots and gaps caused by infectious diseases) as well as living trees that are damaged to varying degrees by pests, diseases or abiotic factors and may remain alive for years. It has been documented, for example, that tree mortality in wind-damaged stands is highly variable [43,44,45,46,47] and that many trees survive the event even if they are severely damaged [48]. This suggests that the inclusion of trees removed from the stand during sanitary cuttings may lead to an overestimation of mortality rates.

In times of global change (climate, air pollution, eutrophication of habitats) and the resulting changes in the basic environmental conditions for tree growth, the assessment of the health status of forests becomes particularly important to maintain their stability, durability and ability to fulfil various functions: provision of wood, protection of biodiversity, sequestration of carbon dioxide. Therefore, there is a need for reliable information on the direction and pace of changes taking place in the forest environment and on the impact of disturbance factors. In response to this need, the development of monitoring programmes and national inventories have been initiated in many countries around the world. Nowadays, large-scale studies on forest health cover almost 85% of the world’s forest resources and are conducted in over 110 countries worldwide [49]. European forest monitoring, coordinated by the International Co-operative Programme on Assessment and Monitoring of Air Pollution Effects on Forests (ICP Forests), is an example of such a programme [50]. It uses a visual assessment of the crown as the main source of information about the health of a particular tree and the whole stand. The basic health indicator is crown defoliation, whose relationship with the condition of other organs and the physiological status of the tree is well documented in the literature [51,52,53,54,55]. The methodology of ICP Forest monitoring, the annual or periodic repeatability of surveys on the same plots and trees, also enables the identification of dead and removed trees. The applied classification of the so-called tree status (indication of alive, dead, removed for various reasons) provides information on the cause of the death or removal of trees and, thus, also enables the identification of trees that have been removed during thinning or sanitary cuttings [56]. All this makes the application of tree survival analysis possible.

Considering the above, one should conclude that there are several possible approaches to calculating tree mortality in managed forests. Our research objective was therefore to find out whether the method used to determine the mortality rate has a significant influence on the results obtained. We hypothesised that the way in which the mortality rate is calculated has a meaningful influence on the results. The second objective was to determine the most appropriate of the three methods described above for calculating tree mortality in managed stands. We hypothesised that the most appropriate method would be the one that takes into account both the dead trees remaining in the forest and the trees removed from the stand, which are most similar to the dead trees in terms of health parameters such as crown defoliation and damage severity recorded in the last year when both the dead and removed trees were still alive.

## 2. Materials and Methods

### 2.1. Data

The study was conducted based on the results of annual assessments of trees on Level 1 plots of the national forest monitoring in Poland between the years 2009 and 2022. The plots were distributed in a systematic network of 8 km × 8 km across the country’s forest areas. The number of plots with 20 sample trees each was 1923 in 2009. The trees were visually assessed in July and August according to the ICP Forests Manual, Part IV: “Visual assessment of crown condition and damaging agents” [33,56], including the assessment of crown condition and damage occurring on the trees. Because of the non-destructive nature of the observations carried out, no plant or wood samples were taken. Due to the varying number of trees of each species, the analyses were limited to the most numerous ones: *Pinus sylverstris* L.—“pine”, *Picea abies* (L) H. Karst.—“spruce”, *Quercus petrea* (Matt.) Liebl. or *Quercus robur* L.—“temperate oaks”, and *Betula pendula* Roth.—“birch”. Trees removed during final felling were also excluded from the analyses, as they were a priori classified as neither removed nor dead trees. For this purpose, an upper age limit was set for the trees, above which the trees were not included in the calculations. In the case of pine, spruce and birch, it was 80 years and in the case of temperate oaks it was 120 years in 2009. The original cohort of trees consisted of trees found alive in 2009 in Level 1 plots of the national forest monitoring in Poland. A total of 16271 pines, 1284 spruces, 2491 oaks and 3521 birches were included in the analyses (Table 1). The average age of the pines and spruces was the same at 52.5 years, while it was slightly lower for the birches (49.6 years) and higher for the temperate oaks (68.2 years).

### 2.2. Study Design and Statistical Analysis

Two types of analyses were conducted. The first one was carried out to find out whether the method of determining mortality rates has a significant impact on the results obtained and consisted of a survival analysis of pines, spruces, oaks and birches during the period 2009–2022 using the life table method [57]. Changes in number of alive, dead and removed specimens in successive annual surveys were used to perform this analysis. Three following variants of data classification were tested, in which complete observations consisted of dead trees remaining on plots—variant 1, accompanied by all trees removed from plots between successive surveys—variant 2 and accompanied by trees removed in sanitary cuttings (described as removed due to biotic or abiotic reasons as well as removed trees whose defoliation was 65% or more in the year prior to removal)—variant 3. The 65% defoliation threshold was adopted in accordance with the third-highest class of tree defoliation as indicated by ICP Forests manual for crown condition assessment [56]. The four categories of trees were distinguished accordingly, i.e., alive, dead remaining on plots, removed and removed in sanitary cuttings (Table 1). In the variant 3 removed trees except those removed by sanitation cuts were treated as individuals, which left the cohort for reasons other than dieback. The non-parametric Wilcoxon–Gehan test [58] was used to compare the survival curves between different variants. The purpose of this test is to determine whether there are significant differences in survival time between the variants. The null hypothesis of the test assumes that both survival curves are equal. In the original paper by Gehan (1965), which refers to a medical study, it is formulated as “treatment A and B (are) equally effective”, against the alternative hypotheses “treatment A (is) more effective than B” or that “treatment A or B (is) more effective”. The adopted *p*-value was 0.05.

The analysis of tree survival was carried out using the statistical package Statistica 13.3 [59].

The second type of analysis was performed to check which of the distinguished tree categories are similar to each other and which are significantly different. This was intended to help to indicate the most appropriate method of tree mortality calculation in managed stands. Since the dead trees remaining on the plots were included in the calculations of mortality rates for obvious reasons, the question remained whether the removed trees or one of their categories—the trees removed during thinning or sanitary cuts—should be included in these calculations. We assumed that this could be indicated by the similarity between the different categories of trees removed from the plot and the dead trees remaining on the plot. To this end, we used data on crown defoliation and tree damage size, as well as tree age and diameter at breast height (dbh) of the trees, determined according to the ICP Forests methodology [60,61], and recorded in the year prior to the detection of tree removal or death. Severity of damage was defined as the sum of the extents (expressed as a percentage of the organ where the damage occurred) of up to the three main symptoms of damage. In this analysis, data on dead or removed trees within the entire 2009–2022 period were used, while characteristics of alive trees were used from 2022. We used principal component analysis (PCA) biplots generated for each species considering four tree characteristics (age, DBH, defoliation, total damage). The biplot technique enabled the determination of relationships between variables and the detailed description of a multivariate data set [62]. In addition, the differences between the determined tree categories were tested using the Kruskal–Wallis test [63]. This is a statistical rank test that compares the distribution of a variable in populations. The test does not assume normality of the distributions. It is sometimes regarded as a non-parametric alternative to the one-way analysis of variance between groups. The null hypothesis is the equality of the distribution functions in the compared populations. Adopted *p*-value is 0.05. As the distribution of some of the variables (i.e., defoliation and severity of damage) deviated from the normal distribution, a data transformation was performed and the PCA analysis was repeated. However, the results did not differ from those obtained with untransformed data. Therefore, to facilitate interpretation, the PCA analysis for the original data is presented.

The statistical analyses of the PCA and Kruskal–Wallis test were carried out using R software version 4.2.3 [64].

## 3. Results

### 3.1. Survival Analysis

The survival probabilities calculated for variants 1 and 2 as well as for variants 3 and 2 in the period 2009–2022 differed considerably for all analysed tree species (Table 2, Figure 1). The greatest differences between the variants were found for birch and spruce. Between variants 1 and 2 and 3 and 2, they were 29 and 26 p.p., respectively, for birch, and 28 and 22 p.p., respectively, for spruce. For pine, the differences between the survival rates calculated for variants 1 and 2 (22 p.p.) and 3 and 2 (19 p.p.) were somewhat smaller. The smallest differences were found for temperate oaks—16 p.p. between variants 1 and 2 and 15 p.p. between variants 3 and 2. A significant difference between tree survival rates in the period 2009–2022 between variants 1 and 3 was found for spruce (with a difference of 5.9 p.p.), birch (2.7 p.p.) and pine (2.6 p.p.). For temperate oaks, this difference was insignificant (Table 2), which is due to the very low number of trees removed in the sanitary cuttings (Table 1). These results clearly show that the way of treatment of the removed trees has a major influence on the results of the survival analysis.

The lowest probability of survival between 2009 and 2022 was found for spruce—the survival rates for all variants were the lowest among the tree species analysed. In variant 1, it reached 81.6%, in variant 2 it was 53.3% and in variant 3, in which mortality was calculated taking into account the dead trees remaining on the plots and the trees removed in sanitary cuttings, it was 75.7%. The survival rates of birch were similar to those of spruce (84.1%, 55.2% and 81.4%, respectively), while those of pine were significantly higher (93.3%, 71.7% and 90.7%, respectively). The lowest mortality rates and the highest survival in the period 2009–2022 were found for temperate oaks. For variant 1, it was 95.1%, 79.2% for variant 2 and 94.3% for variant 3. The analysed tree species can thus be ranked according to decreasing survival probability in the period 2009–2022 in the following order: temperate oaks, pine, birch and spruce.

### 3.2. PCA and Kruskal–Wallis Tests Results

Principal component analysis (PCA) was performed for the individual tree species to observe the variation and correlation patterns for each data set separately. PCA generated four principal components (PCs), with the first two PCs explaining over 80% of the total variance in the data examined. Biplots for each species illustrating the relationships between the variables and the components showed a positive correlation between age vs. dbh (r > 0.56) and defoliation vs. damage totals (r > 0.66) and no relationship between either pair of factors (|r| < 0.27) (Figure 2). Two pairs of variables showed correlations with two different principal components (PC1 and PC2).

Group-specific concentration ellipses help to recognise similarities and differences in the relationships between the traits in the distinguished categories of trees. Different behaviour of PCA concentration ellipsoids was observed among tree categories. The concentration ellipsoids for dead trees and trees removed due to sanitary cuttings on the one hand and for living trees and trees removed due to thinning on the other hand overlapped, indicating that the variables in these groups have similar patterns of variation and correlations (Figure 2). Regardless of tree species, dead trees and trees removed due to sanitary cuttings had higher mean values for defoliation and damage than living trees and trees removed due to thinning.

This can be seen in the position of the centroids for both classes (shift towards defoliation and damage total vectors) and in the shape of the ellipses, which are elongated for the damage parameters. Small distances between the centroids also indicate a similarity in the characteristics of these two groups of trees (Figure 2). This can be seen particularly clearly in the pine and somewhat less so in the other tree species, which is due to the different number of specimens in the tree categories—in the pine it is many times higher than in the spruce, temperate oaks and birch (Table 1). The results of the Kruskal–Wallis test (Figure 3) also showed that the differences in average defoliation and damage parameters between living trees and trees removed due to thinning, and between dead trees and trees removed due to sanitary cuttings, were not statistically significant for most species. The only exceptions were pine, where defoliation was higher in trees removed by thinning than in living trees, and oak, where damage size values were lower in trees removed by thinning. For the second pair of tree categories (dead trees versus trees removed due to sanitary cuttings), such a significant difference was only found for damage size and spruce—trees removed in sanitary cuttings had lower last recorded values for this parameter than dead trees. All this supports the general finding that the living trees were generally comparable in terms of defoliation and severity of damage to the trees removed during thinning and the dead trees to the trees removed in sanitary cuttings.

## 4. Discussion

The results of the survival analysis confirmed the importance of the type of treatment of the removed trees, either by sanitary cuttings or by thinning, for the calculation of mortality rates. According to Dobbertin [32], it can therefore be assumed that mortality limited to trees alive in one year and dead in the next year results in an “observed” or “minimum mortality” that is lower than the “actual mortality”. This is because actual mortality includes dead trees that were felled before they were recorded, as well as trees that would have died by the next survey if they had not been felled. A similar remark applies to the case of variant 2, in which all trees that were removed between successive surveys were included in the group of dead trees. The reason for this is that the removed trees can also include living trees that would remain alive for many years. Furthermore, in periodic inventories it is practically impossible to estimate the categories of trees removed from the stand and determine the actual mortality on this basis. This means that, in practice, it is not possible to accurately determine the actual mortality of trees in managed stands undergoing intermediate cuttings. Nevertheless, the limits of this mortality can be determined by considering only the dead trees remaining in the stand as mortality (“minimum mortality”) and by including both the dead and the removed trees in the calculation (“maximum mortality”). Although the range between minimum and maximum tree mortality is very large (in our study between 15.9 and 28.9 p.p., depending on the tree species), which emphasises the uncertainty in this matter, it seems reasonable to determine it by applying survival analyses when assessing forest health. It could be considered as a kind of “confidence interval” for tree mortality. The approach found in the literature [3,39,40], where only the dead trees remaining in the stand are included in the mortality calculations for managed stands, should therefore be rejected as it is based on incomplete data and leads to an underestimation of mortality rates. The exclusion of all trees removed from the stand, including trees that have already died or would die by the next survey, leads to a misjudgement of the health status of the stand, which could result in no protective measures being taken.

The PCA analyses and Kruskal–Wallis tests carried out were intended to provide information on whether the health characteristics (crown defoliation and severity of damage) of the trees removed during sanitary cuttings and thinning were more similar to the trees found dead the following year and still remaining in the stand or to the living trees. We hypothesised that the similarity between a certain category of removed trees and the dead trees would give an indication of the correctness of the approach that takes into account the removed trees of this category in the calculation of mortality. Furthermore, the similarity with living trees would indicate that the removed trees of a certain category should not be taken into account when calculating the mortality rate. The results of the analyses carried out were clear—the trees removed during thinning were similar to the living trees in terms of defoliation and severity of damage, while the trees removed in sanitary cuttings were more similar to the trees found dead the following year. These results argue in favour of including the trees removed during sanitary cuttings in the calculation of mortality rates, as Czabańska et al. [41] and Grodzki et al. [42] have done. However, considering on the one hand that living trees (damaged or in neighbourhood of infestation spots) are also subject to such cuts [45,46,47], it can be concluded that mortality rates calculated in this way might be overestimated. On the other hand, thinning, the main cause of tree removal, leads to a lower number of trees in the stand and consequently to lower tree mortality as there is less competition between trees [65,66]. This is particularly evident in young stands, which are characterised by an initially high tree density [67]. As thinning improves the physiological and growth-related response of trees to stress factors, it is therefore likely that sanitary cuttings in managed stands are much less intensive than natural dieback in the same stands excluded from forest management [68,69].

The comparison of oak health assessment based on evaluation of tree defoliation [70] and tree survival analysis presented here serves as an example of the complementarity of these two methods. Their results were fundamentally different. According to the first method, temperate oaks in Poland showed the highest leaf loss among the main forest-forming tree species in the period 2012–2021, indicating their poor health status. The second method, survival analysis, carried out for the period 2009–2022, showed that oaks had the lowest mortality rates, both minimum and maximum, among the species studied, which speaks in favour of their good vitality. This may indicate that health of oaks in recent years has not been as poor as the defoliation assessment suggests. The opposite example is spruce, a tree species that has the highest level of defoliation and damage among conifers [70] and has the highest mortality rate. Both cases argue in favour of routinely examining tree mortality/survival in large-scale forest health studies.

## 5. Conclusions

Survival analysis is an objective method that provides verifiable results and makes it possible to examine temporal and spatial trends and relate them to changes in environmental factors affecting forest ecosystems. That is why it seems reasonable to routinely use it in Europe in long-term and large-scale studies of forest condition as a complementary indicator of the health of managed forests, apart from crown defoliation, the severity of damage or tree growth.

The results of the present study, regardless of tree species analysed, show remarkable differences in tree mortality rates for 2009–2022 in managed forests in relation to the way removals were treated in the analysis. As it is not possible to determine whether the removed trees were dead or alive at the time of felling and whether they would have survived until the next survey, it is not possible to determine the actual tree mortality in the managed forests. Nevertheless, with the help of survival analysis, it is possible to determine the upper and lower limits between which it lies. These limits are determined by including in the mortality calculations the trees that died in a given period and remained in the stand (“minimum mortality”) or both the dead trees and all trees removed from the stand between the surveys (“maximum mortality”).

The difference between minimum and maximum mortality is very large. One way of limiting this difference could be to include in the mortality calculations also trees that were removed as sanitary cuttings. The results of the PCA and Kruskal–Wallis tests showed similarities between dead trees and trees removed in sanitary cuttings in terms of crown defoliation and severity of damage, which speaks in favour of the appropriateness of this approach. However, it was found that the main cause of tree removal is thinning, which leads to a greater sparsity of trees in the stand and consequently limited natural dieback due to reduced competition between trees.

Further research is needed to determine the relationship between the thinning intensity and the decrease in tree mortality rates caused by changes to the natural dynamics in the stand.

## Figures and Tables

**Figure 1 plants-13-00248-f001:**
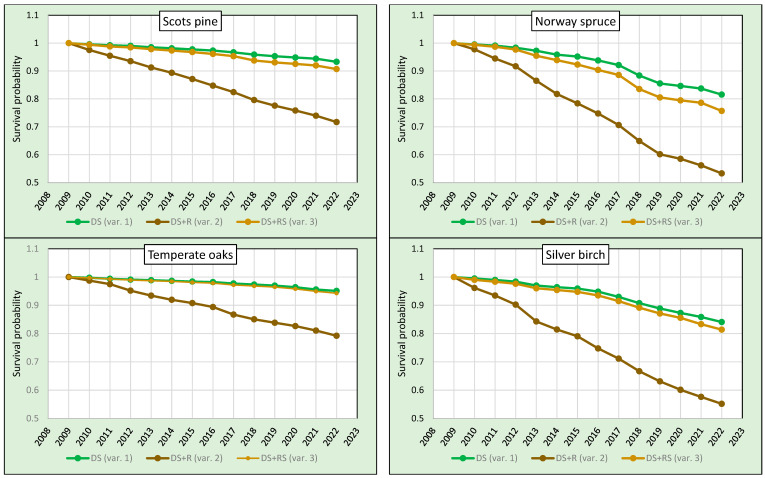
Survival of pines, spruces, oaks and birches in the period 2009–2022.

**Figure 2 plants-13-00248-f002:**
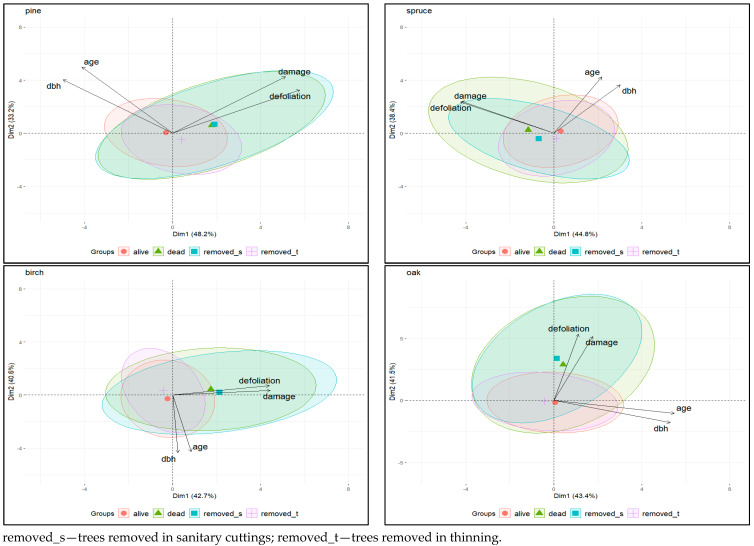
PCA biplots of the first two principal components for analysed tree species.

**Figure 3 plants-13-00248-f003:**
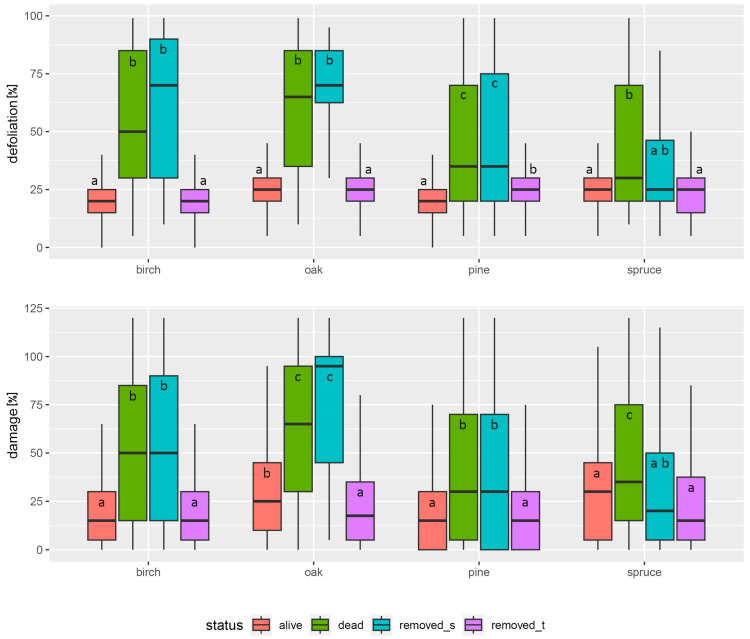
Box-and-whisker plots of all tree species characteristics for different tree categories. The first and third quartiles define the box, the median is shown as “—”, and the whisker defines the range of the data without outliers. Letters “a”, “b”, “c” identify homogenous groups with no significant differences (based on the Kruskal–Wallis test).

**Table 1 plants-13-00248-t001:** The number of trees by species and status category in the year of beginning or end of analysis or within the period 2009–2022 and its relative value to the 2009.

Category of Tree	Year/Period	Units	Scots Pine	Norway Spruce	Temperate Oaks	Silver Birch
Alive	2009	Number	16,271	1284	2491	3521
		% of 2009	100	100	100	100
Alive	2022	Number	11,767	707	1982	1997
		% of 2009	72.3	55.1	79.6	56.7
Dead	2009–2022	Number	918	177	108	419
		% of 2009	5.6	13.8	4.3	11.9
Removed in thinning	2009–2022	Number	3202	332	382	1023
		% of 2009	19.7	25.9	15.3	29.1
Removed in sanitary cuts	2009–2022	Number	384	68	19	82
		% of 2009	2.4	5.3	0.8	2.3
Removed total	2009–2022	Number	3586	400	401	1105
		% of 2009	22.0	31.2	16.1	31.4

**Table 2 plants-13-00248-t002:** Results of the Wilcoxon–Gehan test for the significance of the differences between the survival rates for variants.

Species (Age of Trees)	Results of Wilcoxon–Gehan Test					
	Var. 1 versus Var. 2		Var. 1 versus Var. 3		Var. 2 versus Var. 3	
	Stat. *Z*	*p*	Stat. *Z*	*p*	Stat. *Z*	*p*
Pine (21–80)	49.226	0.00000	8.216	0.00000	−42.568	0.00000
Spruce (21–80)	12.089	0.00000	3.436	0.00059	−12.063	0.00000
Oaks (21–120)	16.315	0.00000	1.244	0.21363	−15.328	0.00000
Birch (21–80)	25.972	0.00000	2.814	0.00490	−23.611	0.00000

## Data Availability

The analysed data sets supporting the results of this study are available on request from the corresponding author.

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
