# Peer review of "“Mortality, or not mortality, that is the question …”: How to Treat Removals in Tree Survival Analysis of Central European Managed Forests"

_plants, 2024, doi:10.3390/plants13020248_

Round 1

Reviewer 1 Report

Comments and Suggestions for Authors

Based on the Polish level I forest monitoring data, the study demonstrates how the inclusion or exclusion of trees removed by humans changes the calculated survival and mortality rates significantly. Different variants allow to determine a range between minimum and maximum mortality. Considering a set of tree attributes, the study further shows the similarity of trees removed during salvage cutting to dying trees, and the similarity of trees removed during thinning to vital surviving trees. Based on these findings, it is suggested to include salvage cuttings, but exclude thinning removals in mortality rate calculations. This research is of high relevance, for monitoring and understanding tree mortality in managed forest ecosystems. However, I have some remarks, which the authors should consider, before the manuscript can be published in Plants.

Major comments:

·         I suggest to rename the variants 2 and 3 according to increasing number of included removal categories. It sounds like a trivial thing, but for me it was very confusing that variant 3 is the one in the middle and I think Table 1 and Figure 1 would be much easier to understand, if the names of var. 2 and 3 were swapped.

·         At several places in the manuscript, I had difficulties relating the numbers given in the text to the results shown in tables and graphics (see also minor comments). Please check them again and make sure to be precise in reporting “percentage points” or “%-points” and not “%” when talking about differences between percentages.

·         The position of the Materials and Methods section between Discussion and Conclusion is confusing. Since the MM are necessary to understand the Results and also the MM is not long, it should be placed at its conventional position between Introduction and Results.

·         I am not familiar with the details of the Polish level I monitoring. According to the ICP manual, in many countries the sample trees have to be in Kraft classes 1-3. If a tree is outcompeted, i.e., degraded to a class > 3, it leaves the sample and gets status 23. I would be curious about how such trees were treated in the study. Of course, such trees don’t die immediately. But could they maybe serve as a proxy for self-thinning? Including or excluding them could maybe be used to create a new variant for approximating “true mortality”. Also, this raises the general question of how biased mortality estimates based on a population of Kraft class 1-3 trees is compared to the population of all trees including the understory. I am not necessarily suggesting to perform new analyses about this topic. I am just curious to read the authors’ opinion about it, and suggesting to discuss this issue.

Minor comments:

·         L. 51: to be precise, it should be stated that “living trees at the end of the assessment period” does not include ingrowing individuals, which were not part of the “living trees at the beginning of the assessment period”.

·         L. 290: 8 km x 8 km

·         L. 126: Please explain what the percentages refer to. 29% and 26% of what? And when talking about differences in rates, do you mean percentage points or actual percent?

·         L. 138: “- 53.3%” and “- 75.7%” reads like minus. Suggestion: replace “-“ with “it was”.

·         L. 140ff: It would be much easier to read, if you would only talk about survival rates here, because the numbers are survival rates. It is confusing if you talk about lower mortality rates for pine, while reporting higher numbers.

·         L. 143: replace “-“ by “with”.

·         L. 146f: Is this a valid conclusion? Isn’t the lower oak mortality also an effect of the species’ intrinsic life cycle properties, i.e., slow growing, large maximum size, long life expectancy? I would say, the stability of the mortality rate over time for each species is an indicator for maintenance of good conditions, but not necessarily the absolute survival rates between species.

·         Fig. 1: for the subplot “temperate oaks”, the grid pattern in the background is not aligned with the data points and years, like in the other subplots. And typo in caption: 202 should be 2022.

·         L. 226: How were the 15.9% and 28.8% calculated? Are they the differences between var1 and var2 after 13 years given in percentage points? If so, then these numbers are not in line with Table 2. Also, it would be very interesting to report the differences between the variants in terms of percentage points of annual mortality rate (e.g., the mean annual rate per species over the 13 years).

·         L. 302: Would it be possible to provide information about the mean age or age class distributions of the four species?

·         L. 322: A brief explanation of the Wilcoxon-Gehan test would be helpful for readers, who are unfamiliar with the method. I think PCA and Kruskal-Wallis test are familiar methods for a broad audience, but survival analysis not so much.

Reviewer 2 Report

Comments and Suggestions for Authors

The document starts with the premise that mortality determination in managed forest is a hard task. Your reference to Dobbertin applies to mortality in slow growing plantations.  Given the title of the article, you should consider that in fast growing plantations, (who are also managed forests), this observations is not correct.  Most often than non, we do know that a tree has died, we just don't know when in between measurements.  I would recommend either expanding your focus or narrowing your title to European forests.  There is a plethora of studies relating mortality (due to its various forms, competition, disease, disturbance a.s.o) to elapsed time, tree size and tree height.  None of those were mentioned in your manuscript.  The mathematical treatment of the test does not allow you to see if you have the same results over time.  I was lost by the figures shown, initially thinking you were only looking at three plots.  There seems to be a disconnection between your objectives and what has been done with respect to mortality evaluation in a mathematical sense in forestry. This seems to be a non-issue for managed stands over the world, and more of a local problem that needs to be addressed accordingly.  The censored regression approach is sound, but one that has been published before.

Reviewer 3 Report

Comments and Suggestions for Authors

General comment

Based on 13 years of survey data on pine, spruce, oak and birch trees in Poland, including the amount of leaf fall and the extent of damage, this paper finds that the tree felling rate is one of the methods to assess forest health, which provides some insights for rational forest management.

Major comment1

1.       The idea of this paper does not seem to be very high, because the tree mortality rate in the conclusion is largely affected by human activities such as logging, so it is not a good assessment of the actual state of the forest.

2.       Unfortunately, there does not appear to be a clear model for calculating mortality and making health assessments.

3.       This paper does not provide a good solution to the instability of calculating mortality, which reduces the reliability of the conclusions

Minor comment1

1.       The different methods for assessing numerical mortality are not clear in the abstract of this paper. There is also a bit of haste at the end, without briefly introducing the results of this study

2.       Managed forest mortality seems to be more scientific to rely on manual statistics by managers? Harvesting for human needs is an important process in managed forests, so what is the point of this article?

3.       It is recommended to use the same font as the annotation in the lower left corner of Figure 2 and the explanation in Figure 2

4.       Part 2.2 is a presentation of the results, but there are many of these presentations that should perhaps be included in the introduction, and it is recommended to condense this section

5.       This paper does not provide a good solution to the instability of calculating mortality, which reduces the reliability of the conclusions

6.       The conclution also seems to be full of speculative theories, and there is no clear conclusion about the method of assessing mortality.

Round 2

Reviewer 3 Report

Comments and Suggestions for Authors

I have no further comments.

Author Response

Dear reviewer,

Thank you for the comment and aproval of our manuscript.

Kind regards,

Paweł Lech & Agnieszka Kamińska